# A Combined Approach for Accurate and Accelerated Teeth Detection on Cone Beam CT Images

**DOI:** 10.3390/diagnostics12071679

**Published:** 2022-07-10

**Authors:** Mingjun Du, Xueying Wu, Ye Ye, Shuobo Fang, Hengwei Zhang, Ming Chen

**Affiliations:** 1Institute of Biomedical Manufacturing and Life Quality Engineering, Shanghai Jiao Tong University, Shanghai 200240, China; alphadu@sjtu.edu.cn (M.D.); zhw_shjd827@sjtu.edu.cn (H.Z.); 2Department of Prosthodontics, Shanghai Stomatological Hospital & School of Stomatology, Fudan University and Shanghai Key Laboratory of Craniomaxillofacial Development and Diseases, Fudan University, Shanghai 200040, China; xueyingwu_kq@fudan.edu.cn (X.W.); yeye_91@126.com (Y.Y.); fsbm1995@163.com (S.F.)

**Keywords:** teeth detection, YOLO v3, cone beam computed tomography, convolutional neural network, metal artefacts

## Abstract

Teeth detection and tooth segmentation are essential for processing Cone Beam Computed Tomography (CBCT) images. The accuracy decides the credibility of the subsequent applications, such as diagnosis, treatment plans in clinical practice or other research that is dependent on automatic dental identification. The main problems are complex noises and metal artefacts which would affect the accuracy of teeth detection and segmentation with traditional algorithms. In this study, we proposed a teeth-detection method to avoid the problems above and to accelerate the operation speed. In our method, (1) a Convolutional Neural Network (CNN) was employed to classify layer classes; (2) images were chosen to perform Region of Interest (ROI) cropping; (3) in ROI regions, we used a YOLO v3 and multi-level combined teeth detection method to locate each tooth bounding box; (4) we obtained tooth bounding boxes on all layers. We compared our method with a Faster R-CNN method which was commonly used in previous studies. The training and prediction time were shortened by 80% and 62% in our method, respectively. The Object Inclusion Ratio (OIR) metric of our method was 96.27%, while for the Faster R-CNN method, it was 91.40%. When testing images with severe noise or with different missing teeth, our method promises a stable result. In conclusion, our method of teeth detection on dental CBCT is practical and reliable for its high prediction speed and robust detection.

## 1. Introduction

In contemporary dentistry, digital technologies such as CBCT, intra-oral 3D scan, 3D printing, and personalized treatment planning play an important role in both research and practice. These technologies hold promise for more predictable, objective, and effective treatment while reducing iatrogenic complications. In the past, diagnostic information was collected through clinical interviews, plaster models, and chair-side observations. The information was then interpreted by experts to derive a treatment plan for implementation, which depended upon doctors’ experience. Today, the new technologies can replace human eyes and hands [1]. By using machine learning to augment the image, diagnosis can be more accurate and objective, and treatments can be personalized. A computer does not get tired from grueling tasks, and it can take in a great deal of data and process the information incredibly quickly. In this study, the computer takes in a set of different CBCT images. Then, once the computer is given data of experts’ diagnoses, it can detect and classify different teeth in the human jawbone. Thus, given a new patient’s CBCT images, the machine can easily match it with patterns that were found in the training set of expert diagnostics.

CBCT examination has been widely used in dental practice. It offers highly accurate volumetric data on jaw bones and teeth, with relatively low radiation doses and cost [2]. CBCT plays an important role in auxiliary diagnosis of oral diseases. Several studies have compared the diagnostic accuracy of CBCT with conventional 2D radiography [3,4,5]. CBCT has been shown to significantly increase the detection rate of tooth root canal spaces and periapical areas for the evaluation of dental infection and pathology compared with conventional imaging [6]. It also suggests that CBCT enhances the recognition of periapical bone lesions and offers improved diagnostic accuracy, treatment planning, and thus, prognostic outcomes. Amongst the most frequent applications are CBCT-guided implant surgery [7], CBCT-guided endodontics and apical surgeries [8], CBCT-based planning and fabrication of donor teeth replicas, surgical guides for successful tooth auto transplantation (TAT) [9], digital orthodontic applications [10], and virtual orthognathic surgery planning [11]. All of the above applications rely heavily on the accuracy of image processing to ensure trustworthy pre-op treatment planning. At present, computer-aid image analysis technology is the main research direction for medical image processing in three aspects: classification, detection, and segmentation.

The existing CT computer-aid software can perform simple image processing tasks such as scaling, cropping and threshold segmentation, but they can hardly provide the location and classification information of each tooth automatically [12]. Tooth location and classification information is usually important for tooth segmentation, making it more accurate [13]. M. P. Muresan’s research found an adherent edge of a tooth on CT can cause inaccurate tooth segmentation, and a detection process was introduced to solve the problem, which boxes teeth into different bounding boxes with only one tooth inside per box [14]. In this case, a teeth detection step is usually necessary for most tooth segmentation tasks. However, this step is challenging for traditional methods, so CNN based detection methods are better candidates for teeth detection tasks. Our goal is to build a system that receives a set of CBCT images and carries out auto layer classification on z-axis and teeth detection. For each tooth we have a tooth bounding box, a classification label, and several layer labels (indicate crown, root, etc.). Compared with other methods, our approach makes it easier for humans to perform labeling and has a high speed for training and prediction.

## 2. Related Works

In the field of dental informatics there are many approaches developed for dental diagnosis using different types of radiographic images such as bitewing, periapical, panoramic images and CBCT images. Accordingly, different scholars have made different explorations of teeth detection and classification with different data formats. Some methods are based on traditional feature extraction methods such as the contour detection method [15], level set method [16] and graph-based method [17,18] which calculated the similarity of different images, which include teeth for person identification. Traditional features, including region and contour information of teeth [19], the Fourier descriptors of teeth contours [20], and multiple criteria such as area/perimeter ratio and width/height ratio [21], were gradually applied to teeth detection in forms of dental bitewing and periapical X-rays. These methods use hand-designed features which require less labeling efforts, but the traditional feature extractors can be easily influenced by the variance in X-rays.

In the last few years, the popular CNNs such as VGG16, Resnet and Densenet, which have achieved great success in natural image applications, have realized many clinically significant applications in medical images. Miki et al. [22] investigated the application of a deep convolutional neural network (DCNN) of AlexNet network architecture for classifying tooth types on dental CBCT images. Zhang et al. [23] proposed a special label tree for teeth detection in dental periapical X-rays before numbering the teeth according to some rules. With the emergence of better CNN architectures, some studies [24,25,26] focus on the combination of Faster RCNN [27] network and rule-based modules for the detection and refinement of teeth sequence. Zhang et al. [28] applied a special label construction technique to decompose the teeth classification task and used a multi-task CNN to classify the teeth positions with a proposal correlation module, which utilizes the information between teeth positions. These methods have achieved good performance in their tasks. However, most methods are not designed for dental CBCT images, and their applications are restricted to clinical practice due to the limitations of computing hardware and the inefficiency of algorithm performance.

To improve the robustness and effectiveness of teeth detection and classification, this paper proposes a two-stage teeth detection procedure composed of a teeth classification module based on a redesigned mini-VGG network and a teeth detection module based on YOLO v3 network. In order to leverage the prior spatial knowledge in CBCT images, we use YOLO v3 network to detect teeth hierarchically with auxiliary procession of the proportion-based division. The experimental results show that our two-stage framework achieves equivalently good precision of teeth recognition compared to prior research, with a significant increase in speed and huge moderation for computing resources.

## 3. Dataset and Relevant Knowledge

The CBCT (NewTom, Italy) used in this study offered a cylindrical volume of reconstruction up to 15 × 15 cm with a 16-bit gray density and 0.3 mm voxel size under the setting of 110 kV tube voltage and 3.6 s exposure time.

All acquired data were saved and exported in Digital Imaging and Communications in Medicine(DICOM) format. A rapid review by the ethics committee was adopted for exemption from informed consent and all the data used with de-privacy processing so that the privacy of patients and the confidentiality of identity information are guaranteed.

A total of 25 dental CBCT scans were used in this study. Twenty CBCT scans were randomly collected for dentitions of minor defect (no more than two missing teeth) with no continuous missing teeth. Among these CBCT scans were 10 scans with no wisdom teeth, 6 scans with one missing tooth, and 1 scan with two separate missing teeth. The patients were aged from 20 to 52 years. Another five scans were used for one special case study, which will be discussed in Part 6 for situations of continuous missing teeth and severe metal artefacts. Patients were aged from 71 to 77 years. The training samples are different slices from different patients. The original intention of our study is to enhance the 2D images of CBCT scan before 3D reconstruction so that we will be able to perform segmentation of teeth in our further study from the beginning of image processing.

To identify every tooth with a distinct label, the Federation Dentaire International system, FDI tooth numbering system was used (Figure 1a).

In this study, the dental arch was divided into five blocks in Figure 1b based on:The incisors: teeth 22, 21, 11, and 12 of the upper jaw or teeth 32, 31, 41, and 42 of the lower jaw;The right canines and premolars: teeth 13, 14, and 15 of the upper jaw or teeth 43, 44, and 45 of the lower jaw;The left canines and premolars: teeth 23, 24, and 25 of the upper jaw or teeth 33, 34, and 35 of the lower jaw;The right molars: teeth 16, 17, and 18 of the upper jaw or teeth 46, 47, and 48 of the lower jaw;The left molars: teeth 26, 27, and 28 of the upper jaw or teeth 36, 37, and 38 of the lower jaw.

This kind of division is the best one during our experiments. If blocks are too big, teeth belonging to other blocks are often included in the box. If the blocks are too small (even one tooth per block), the detector can not detect blocks well when some teeth are missing. This kind of division is also based on the layout of human teeth. To train the detector, doctors are required to label rectangle boxes bounding these blocks with the software labelImg.

## 4. Methodology

The whole detection networks are organized as Figure 2. First, we use a classification network to perform pre-processing, which divides a set of CBCT images into different layer classes. In the second stage, an object detection network extracts the main area (ROI) then predicts the bounding box (we call it the detected bounding box) of blocks of teeth. With proportion-based division, we obtain the tooth bounding box and finish the teeth detection procedure.

### 4.1. Pre-Processing

In this stage, we input a set of CBCT images decoded from dicom files. Then, each image passes through the CNN network and a switch in Figure 3. The output are images which have been divided into five classes. (Lower no-tooth area, lower crown area, overlapping area, upper crown area, and upper no-tooth area).

Previous studies have proved CNN networks such as AlexNet a good example to which to apply tooth classification, while our first task is also dividing CBCT tooth layers into different classes [22]. The output channel has three classes (corresponding to no-tooth area, crown area, and overlapping area). The hyperparameters are set as learning rate $1\times 10^{-5}$, batch size 32, exit at epoch 50 or when validation accuracy stabilizes for 10 epochs, input image size 128 × 128 × 1. The training platform contains a Nvidia RTX 3080Ti 8G. We tested popular models: AlexNet, VGG-16 and ResNet 50. The result shows that in this CBCT layer classification task, VGG-16 performs the best with the least overfitting phenomenon.

According to our task, we redesigned a mini-VGG model based on VGG in Figure 3. After the same experiment above, our mini-VGG model showed similar performance as VGG-16, while the training speed improved. This model receives a gray image with 128 × 128 size. The network uses random initialized weight; the loss function is a cross-entropy function for a common classification network [29]. The hyperparameters are the same as above. In the output stage, we extend three classes into five classes (illustrated above). The theory behind this is that we assume CBCT is scanned from bottom to top, so class labels change in order, e.g., when we observe the class label change from no tooth to crown, it means the layer label should change from lower no-tooth area to lower crown area.

Another problem in practice is misclassification. With a classification accuracy higher than 95%, there are still chances that mistakes will occur. Once a mistake triggers a class label change, the current layer label will be switched too early. To handle this, we use an anti-noise switch algorithm: when a layer’s class label changes to a new one, only if two or more following layers continuously hold the same new class label, the current layer label can switch to another state. With 95% accuracy, the probability of two continuous mistakes occurring is 0.25%, whereas for three continuous mistakes occurring it is 0.0125%, which is enough for enhance accuracy.

### 4.2. Teeth Detection

In this stage, we apply teeth detection and divide each tooth by rectangle bounding boxes. Most researchers use two-stage detection networks (e.g., Faster R-CNN) and bounding boxes are labeled and trained per tooth [14,24,30]. Our method makes a major difference to the detection network and detection bounding boxes. The first change is that we use the one-stage detection network YOLO v3 rather than a two-stage detection network. This network is seldom used in clinical CT detection because one-stage networks have lower accuracy on small object detection, and usually objects on CT are small. However, the YOLO-based network trains and predicts much faster than two-stage methods such as Faster R-CNN [31]. To improve performance, we use multi-level detection. (1) We detect the main area, a box bounding all teeth tightly; (2) we detect blocks of teeth as mentioned in Section 3. These objects are large enough according to the anchor size in Redmon J’s paper [32]. A comparison between a tooth detection box and middle size anchor box is shown in Figure 4.

YOLO detector training hyperparameters are as follows: learning rate $1\times 10^{-4}$, batch size 10, input image size 416 × 416 × 3, train for 100 epochs.

Another difference is in the bounding box. One tooth in one box often causes problems. Dentition with missing tooth or dental restorations with metal artefacts cannot be well detected, causing disorder of tooth bounding box labels. On the other hand, many teeth are similar in shape, so detection networks have low accuracy when distinguishing these teeth [24]. Here, we propose a combination of teeth detection and proportion-based division to generate each tooth’s bounding box. We detect five detected bounding boxes for each block. Figure 1b. In the detected bounding box, we locate the tooth bounding box based on proportion. According to Section Formula, a proportion located point inside a detected bounding box has a coordinate:(1)$$x=\frac{x_{1}+\lambda_{x}x_{2}}{1+\lambda_{x}}$$
(2)$$y=\frac{y_{1}+\lambda_{y}y_{2}}{1+\lambda_{y}}$$

Here, $\left(x_{1},y_{1}\right),\left(x_{2},y_{1}\right),\left(x_{2},y_{2}\right),\left(x_{1},y_{2}\right)$ for bounding box’s vertices have coordinates, (x,y) for the target point’s coordinate. $\lambda_{x}$ for the position ratio and $\lambda_{x}=\left|\frac{x-x_{1}}{x_{2}-x}\right|$, the same as $\lambda_{y}$. Usually people use the left-top and right-bottom points’ coordinates to describe a bounding box, a coordinate (L,T,R,B) corresponding to $\left(x_{1},y_{1},x_{2},y_{2}\right)$ above. Taking tooth 45 as an example, the given information is side3R big detected bounding box’s coordinate $\left(L_{0},T_{0},R_{0},B_{0}\right)$. We assume $\lambda_{x}=\frac{1}{6},\;\lambda_{y}=\frac{2}{9}\;$ for the tooth bounding box’s left-top point, $\lambda_{x}=\frac{5}{6},\;\lambda_{y}=\frac{4}{9}$ for the tooth bounding box’s right-bottom point. To optimize lambdas, we chose a CBCT slice from 14 patients. First, we initialized the lambdas by observation. Then, we compared IOU (Intersection over Union) of the generated tooth bounding box and ground truth tooth bounding box and adjusted the lambda values until the mean IOU results were above 85%. We also enlarged each box a little to fit the rotation of scan and malocclusion. Finally, we obtained tooth 45’s bounding box: (3)$$\begin{array}{c} R_{5}:\left(L,T,R,B\right)\\ L=0.17R_{0}+0.83L_{0}\\ T=0.28B_{0}+0.72T_{0}\\ R=0.83R_{0}+0.17L_{0}\\ B=0.72B_{0}+0.28T_{0} \end{array}$$

All tooth bounding boxes are generated like this. The final output is Figure 5.

In conclusion, the whole teeth detection procedure is as follows: For each CBCT set, all images are classified into 5 categories in pre-process stage. From images in the upper crown area, we select the 10th and the last 10th image (images on red and blue lines in Figure 5b), the same for lower crown area. The selected four images are then sent into YOLO v3 detection network and detected main areas. In the main area, use YOLO again to detect five big boxes corresponding to five blocks. Then, we apply proportion-based division to get each tooth’s bounding box. Neighboring CBCT images share these tooth bounding boxes, as is shown in Figure 5b.

## 5. Experiment and Results

In this section, we evaluated the average performance of teeth detection and its performance on severely noised images. In the teeth detection part, we used several metrics to evaluate the detection network. The metrics are mAP (mean average precision), precision, recall, and F1 score. These metrics are tested on the 112 images above. The mAP metric is based on detecting precision and recall, which are frequently used to evaluate the performance of object detection [33]. In object detection, each detected box has four possible classes: True positive (TP), True negative (TN), False positive (FP), and False negative (FN). Whether a box is negative or positive is based on the score each box obtains (usually a box is positive if the score is higher than a certain threshold). Whether it is true or False is based on compliance with the ground truth. Precision is defined as:(4)$$precision=\frac{N_{TP}}{N_{TP}+N_{FP}}$$
Recall is defined as:(5)$$recall=\frac{N_{TP}}{N_{TP}{\;+\;N}_{FN}}$$

$N_{TP}$ stands for the number of True Positive boxes, $N_{FP}$ for the number of False Positive boxes, and $N_{FN}$ for the number of True Positive boxes. *F*1 score is defined as:(6)$$F1=\frac{2\cdot precision\cdot recall}{precision+recall}$$

The Faster R-CNN method is often used in clinical image detection, as is mentioned in Section 4.2. For this reason, we trained a Faster R-CNN detector based on another set of teeth and labels, and these labels (tooth bounding boxes) are on each tooth. The detector directly detects each single tooth. We refer to this method as the plain Faster R-CNN method. We also collected 112 dental CBCT images and three doctors were assigned to perform labeling work on them, respectively. These images never appeared in training sets. The three doctors chosen had different clinical experience. The years of working experience were 15, 10 and 5 years, respectively. On each image we focused on eight teeth (from wisdom tooth to central incisor, named from ×8 to ×1). The doctors labeled rectangle tooth bounding boxes as ground truth with the software labelImg. The sets are defined as test sets. All test sets are illustrated in Table 1.

In an experiment, two detection methods separately detected teeth on each test set. Then, we compared the results of eight desired teeth with ground truth in the test set. The results are illustrated in Table 2. The results show there is no difference in mAP between our method and Faster R-CNN methods. (*p* = 0.835) There is a significant difference in Precision, Recall, and F1 between our method and Faster R-CNN methods. (*p* < 0.001, *p* = 0.034, *p* < 0.001).

We also tested OIR (Object Inclusion Ratio), a common metric in teeth detection. It describes whether a box completely includes an object [30]. OIR is defined as follow:(7)$$OIR=\frac{A_{M}\cap D_{M}}{A_{M}}$$

$A_{M}$ stands for the object actual area for object M. $D_{M}$ stands for the detected box labeled M. If such a detected box does not include any of object M, the object’s OIR equals zero. If a tooth is missing in an image, the tooth is skipped. The OIR of a CBCT image is the mean value OIR of all teeth. A segmentation dataset is required for this experiment, and the ground truth is segmentation masks on eight teeth. Again, the three doctors accomplished segmentation label work on the 112 images above together. We evaluate OIR on each image, and get 112 × 2 OIR values. The results and statistic analysis are in Table 3.

Table 4 presents the mean and standard deviation values between our method and Faster-RCNN methods. There is a statistically significant difference between our method and Faster-RCNN methods (α = 0.05, *p* < 0.01). Additionally, in Lee’s research [13] improvement of OIR between different methods was 4–10%, while ours is 5.8%. This prove a remarkable improvement.

These results show that our method has higher scores in general over plain Faster R-CNN method. The significant advantage over the plain Faster R-CNN method is on precision, because there are a few phenomena in which the detector confuses two teeth. Meanwhile, the recall rate is slightly lower than Faster R-CNN method. The two differences exist because Faster R-CNN method generates a number of possible boxes for one tooth which are not accurate, but the ground truth boxes must be included in some of these boxes. Of course, many of these boxes are wrong. Then, $N_{FP}$ is much larger then $N_{FN}$. In teeth detection, confusion of two teeth is harmful; in this case, the plain Faster R-CNN method is not suitable for clinic use. The precision sometimes is equal to recall in our method, since FP is equal to FN. Our method only predicts high score boxes per tooth. Sometimes a box is near to the object tooth but is not accurate, with a slight shift from the tooth’s center. Then the predict box is FP and the box on ground truth is assigned to FN. Further discussion can be found in Section 6.

We also compare the training cost and prediction speed of two methods. The batch size is 16 and the train set size is 300 images.

The comparison in Table 5 shows that our detection method is faster than the Faster R-CNN method. This advantage will be significant in big data processing, with high detection speed and low requirements for devices.

Our method is robust to severely noised images. Figure 6a,b show detection on a noised CBCT image under our method and Faster R-CNN method.

Metal artefacts (tooth 11 and 22) and overlapping (tooth 28) appear in this CBCT image at the same time. However, our combined detection has immunity to such phenomena. A slight mistake occurs at tooth 13 because of the ambiguous edge. However, in the Faster R-CNN method, the shape of a tooth affects the detection. Mistakes occur at tooth 16, 26 since it confuses 16 and 17, 26, and 27. Tooth 27, which should be 28, is not accurate, since the disappearing part of the tooth caused by aliasing is not included into the box. In addition, the metal artefacts hinder 12, 11, 21, 22 teeth detection, leading to a messy result.

## 6. Analysis and Conclusions

Our proposed method first extracts different tooth regions using a classification network and detects ROI using YOLO v3. All these works promise a better performance for teeth detection on single-layer CBCT. The combined detection method is based on prior knowledge from doctors, making detection more accurate. High prediction speed and robust detection are the main characteristics of our method. The metrics tables above indicate that our method has high precision and a better average performance than simple object detection networks.

Compared with a plain object detection method, our method promises stable detection even in severely noised images. An absence of a tooth does not affect the result because the absence location will be predicted by combined detection, and the number of teeth will not be affected. The higher prediction speed and lower requirements of devices make it possible to deploy in more applications of clinical scenarios.

There are still improvements left for the method. First, proportion-based tooth location suffers from malocclusion or failing to position the patient’s head at standardized orientation during CBCT scanning. (Figure 7a,b) Additionally, sometimes the adherent edge will also cause detection deterioration.

Second, certain circumstances when continuous teeth loss is more than 3 units or there are extreme metal artefacts might hinder the accuracy of teeth detection. (Figure 7c) In this case, manual adjustment would be needed to calibrate the tooth labeling. Instead of labeling manually, tooth by tooth, our method made it possible to select a large detected bounding box of a group of teeth, which saves a lot of work performed by expert experience.

Third, the tooth bounding box is usually larger than the ground truth box to promise a higher including rate, but it still can be smaller. A smaller tooth bounding box makes tooth segmentation more accurate. Therefore, we might need a new way to evaluate box accuracy while slightly adjusting the box boundary. A lot of training is required to obtain a high-quality intelligent diagnostic model for medical imaging.

For further improvement, prior knowledge such as tooth bounding box proportion inside a block box might be encoded into the CNN network and trained together with the tooth detector in order to improve detection OIR and mAP, with better immunity against noise.

At present, there are various medical image information systems being used in different medical institutions. However, these systems are relatively independent from each other, which leads to inconvenience in usage of the radiographic data universally [34]. Dental images are of great importance in dental records collection. Amongst all methods, CBCT examination provides the most comprehensive patient information It is fundamental for us to include 3D dental information into electronic medical recording system for better usage of clinical examination data. Tooth detection is the basis of dental diagnosis. When artificial intelligence is applied to study bulk diagnostic images in dentistry, accurate automatic tooth detection is undoubtedly the first step. Our study established a starting step for automatic tooth segmentation based on original CBCT images (DICOM format) so that the 3D rebuilding would be able to be combined in the patient’s recording system regardless of the different imaging systems for further research.

## Figures and Tables

**Figure 1 diagnostics-12-01679-f001:**
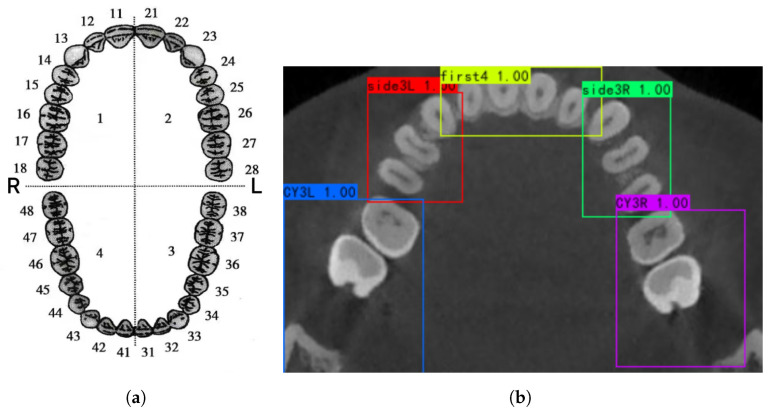
Definition of tooth number and tooth blocks in this research. (**a**) FDI tooth numbering system; (**b**) 5 blocks of teeth for detection.

**Figure 2 diagnostics-12-01679-f002:**
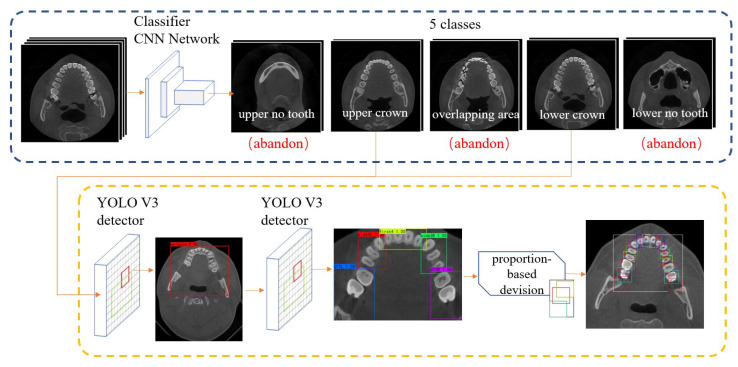
Whole detection network. In the pre-processing part on the left, the input CBCT images are divided into 5 layer classes. Selected images from 2 layers are input into detection part. After multi-scale detection and proportion-based division we get tooth bounding boxes.

**Figure 3 diagnostics-12-01679-f003:**
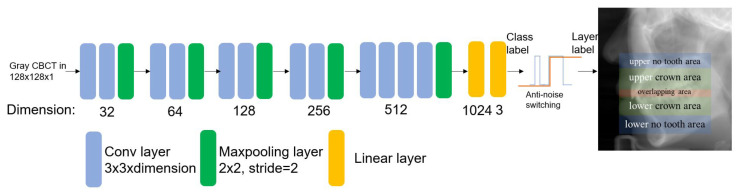
Pre-processing network structure. The input 128 × 128 × 1 gray image passes through our mini-VGG network and gets the class label, and an anti-noise switch determines its layer label.

**Figure 4 diagnostics-12-01679-f004:**
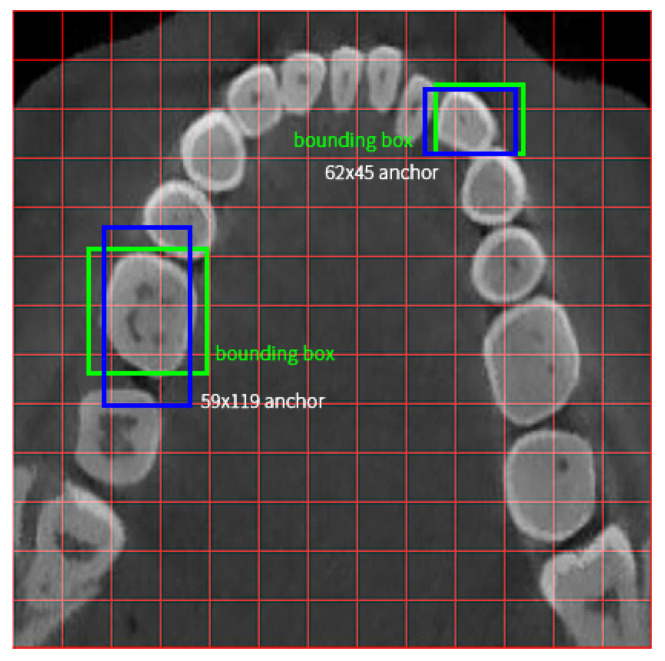
A comparison between tooth detection box and middle-size anchor box. In a main area, each red grid shapes 32 × 32 voxels, and there are a total of 13 × 13 grids. The blue rectangles are default anchors on image after 16× down sample. The green rectangles are middle-sized ground truth boxes. They are generated by the blue anchors on 16× down sample feature map [32].

**Figure 5 diagnostics-12-01679-f005:**
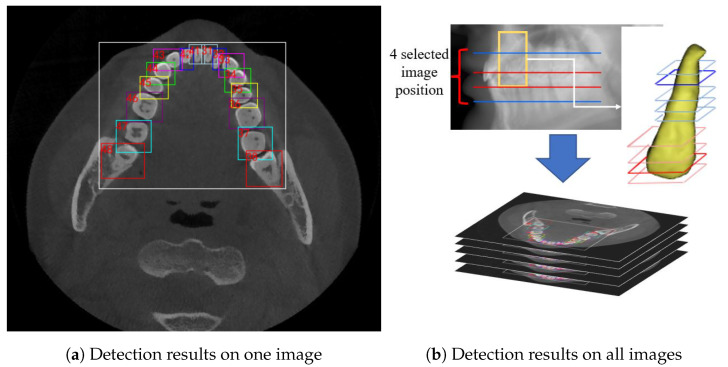
(**a**) shows final detection result on a CBCT image after finishing the whole teeth detection procedure; (**b**) shows how we get tooth bounding boxes in all images: selected layer images use our detection method and detect several sets of tooth bounding boxes (red and blue ones). Then, neighboring images share these boxes.

**Figure 6 diagnostics-12-01679-f006:**
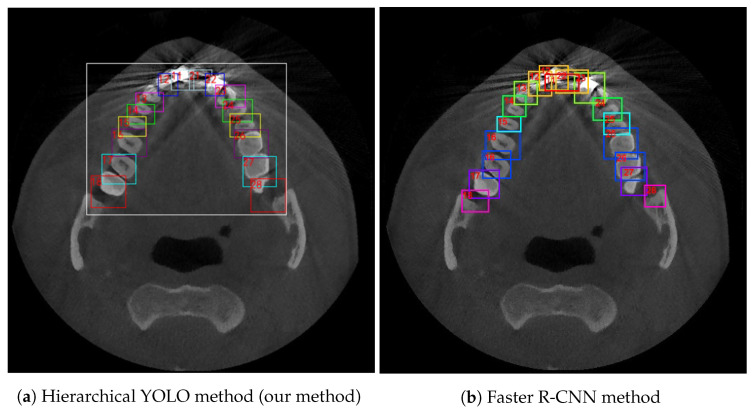
Visualization of the detection results on the severely noised image (**a**) hierarchical YOLO method deals with metal artefacts (tooth 11 and 22) and overlapping (tooth 28) well, (**b**) Faster R-CNN method makes a wrong and disordered detection on several similar teeth and metal artefacts.

**Figure 7 diagnostics-12-01679-f007:**
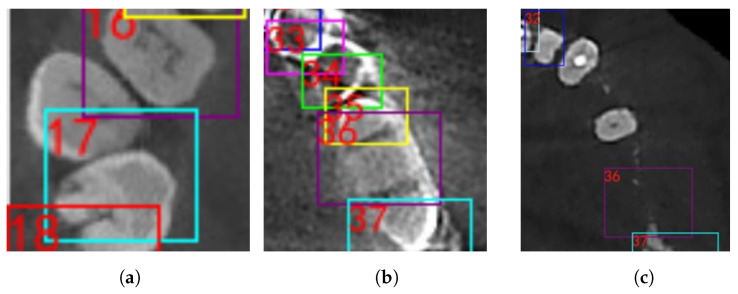
Typical situations of error occur in the detection process. (**a**) Severe malocclusion; (**b**) extreme metal artefacts; (**c**) continuous missing teeth.

**Table 1 diagnostics-12-01679-t001:** Three test sets in the experiment and their contents.

Number	Contributor	Description
1	doctor 1	8 teeth on 112 images
2	doctor 2	8 teeth on 112 images
3	doctor 3	8 teeth on 112 images

**Table 2 diagnostics-12-01679-t002:** Quantitative results of hierarchical YOLO detection method(Our method) and Faster R-CNN detection method on test sets.

	Our Method					Faster R-CNN					*p* Value
	1	2	3	Mean	SD	1	2	3	Mean	SD	
**mAP**	88.77%	81.68%	73.91%	81.43%	12.01%	85.17%	88.08%	71.88%	82.03%	13.43%	0.835
**Precision**	90.82%	86.28%	82.42%	86.15%	10.18%	60.75%	60.73%	55.17%	86.15%	13.35%	<0.001
**Recall**	90.80%	86.19%	82.08%	85.95%	9.75%	94.58%	94.58%	85.40%	90.98%	7.41%	0.034
**F1**	0.9	0.86	0.82	0.86	0.1	0.73	0.73	0.66	0.71	0.12	<0.001

**Table 3 diagnostics-12-01679-t003:** A comparison of two methods on average OIR results and statistic metrics.

Method	Average OIR	N	SD	SE
Faster R-CNN	91.40%	112	0.09832	0.00929
Our method	96.27%	112	0.03946	0.00373

**Table 4 diagnostics-12-01679-t004:** Power analysis between our method and Faster-RCNN method.

Mean	SD	SE	α = 0.05		t	df	Sig.
			Down	Up			
0. 04866	0.1003	0.00948	0.02987	0. 06744	5.134	111	0

**Table 5 diagnostics-12-01679-t005:** Quantitative resource consumption and speed of two methods.

Method	Training VRAM Consumption	Time Consumption per Training Epoch	Time Consumption per Predicting
Faster R-CNN	15 GB	53 s	274 ms
our method	9.5 GB	20 s	53 ms

## Data Availability

The data presented in this study are available on request from the corresponding author.

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
