# Peer review of "A Combined Approach for Accurate and Accelerated Teeth Detection on Cone Beam CT Images"

_diagnostics, 2022, doi:10.3390/diagnostics12071679_

Round 1

Reviewer 1 Report

The research is well designed and carried out, and it is very actual and up-to-date.

Abstract: it is a good summary of the paper, and it it well organized.

Introduction contains enough background informations and adequate references.

Regarding methodology, could you clarify your selection of ground truth?

Is it just the green rectangle of Figure 4 or is there any manual segmentation and a mask create with another software used as reference?

I suggest to consider the following paper in order to extend your work.

PMID: 

  • 34553817

Regarding Conclusion section, I appreciate that you stated the drawbacks of your research. Please add clinical implication and future perspectives of research at the end of it.

Author Response

Regarding methodology, could you clarify your selection of ground truth? Is it just the green rectangle of Figure 4 or is there any manual segmentation and a mask create with another software used as reference? Thanks for remidning. We have add description about ground truth of trainging images in section 3 (line 110), which labels 5 blocks. And description of ground truth when evaluating the performance in section 5(line 207). In evaluation part we use rectangle box for mAP,F1... test, and use tooth mask for OIR test(line 220). (Add: line103: For training the detector, doctors are required to label rectangle boxes bounding these blocks with software labelImg. line171: The doctors label rectangle tooth bounding boxes as ground truth with software labelImg line 180: A segmentation dataset is required for the experiment, and the ground truth are segmentation masks on 8 teeth. )
Please add clinical implication and future perspectives of research at the end of it. In last answer we explain this work is part of detection+segemention, so the main purpose of detection is to support segemention. (line 293) For other clinical implication, we mentioned the method is helpful for electronic medical recording (EMR) system construction. Improvement of our method have been added near line 282
For further research As you mentioned, a detection+segemention work would be more meaningful. Here the article is an achievement of stage and further work is on the way. An compulsory article is required by our institute. What's more, we find such jobs detection+segemention lack detailed detection method and experiment, so we proposed this article.

Revised file in attachment

Reviewer 2 Report

01

“And Object Include Ratio (OIR) metric of our method is 96.55% while Faster R-CNN method is 91.78%.”

No test was performed in order to verify whether this difference was significant or not.

02

Please discuss the costs implicated in the method, in comparison to the ones you are comparing with in the study.

03

How were the 3 doctors chosen for the experiment? Nothing is explained about it. Which level of experience did they have? There is a problem if they presented different levels of experience. Were the doctors calibrated and how?

04

No power analysis was performed. Therefore, it is not possible to know whether the analysis of the results of the present study is a true finding or a pure chance. This may compromise the entire validity of this study.

Author Response

1.“And Object Include Ratio (OIR) metric of our method is 96.55% while Faster R-CNN method is 91.78%.”

No test was performed in order to verify whether this difference was significant or not.

First, we carefully checked and ensured these two values are actually 96.27 and 91.40, since missing teeth should be removed when calculating OIR ( formula in [30] only consider existing tooth, but we also predict missing tooth's box)

Second, we evaluate the improvement with power analysis and compare with results in other paper(line 224).

2.Please discuss the costs implicated in the method, in comparison to the ones you are comparing with in the study.

The main cost is training time and predicting time. We think table 5 Quantitative resource consumption and speed of two methods and its analysis might answer the question?

3.How were the 3 doctors chosen for the experiment? Nothing is explained about it. Which level of experience did they have? There is a problem if they presented different levels of experience. Were the doctors calibrated and how?

The 3 doctors are chosen of different clinicl experience. The years of working experience are 15, 10 and 5 years respectively. (Added in line 204) As for teeth dectection on CBCT images, there are little differences between different observers manually. In our study, we compared two ways of detection methods(herarchical YOLO and Faster R-CNN) with doctors' labels respectively and then we run a statistical analysis to calibrate the influence of human factor.

4.No power analysis was performed. Therefore, it is not possible to know whether the analysis of the results of the present study is a true finding or a pure chance. This may compromise the entire validity of this study.

To prove our improvement compared with Faster-RCNN, power analysis are added in table 2 and 4.

Revised file in attachment

Reviewer 3 Report

The manuscript describes the important problem of identifying the teeth on the CBCT scan. The matter is important and has implications in the dentistry practice. Unfortunately, I found the manuscript not good enough to be published in its current form. The most important remarks are presented below.
1. The introductory part (including Related Works) would benefit from short information on the application of CBCT in disease diagnostics.
2. All mathematical symbols used must be explained! (e. g. lambdas in eq. 1.2)
3. The term pixel is not appropriate for 3D images as CBCT, use voxel instead.
4. Are the voxels cubic in shape? Please, give info in the manuscript.
5. Provide more information on the used images dataset (at least the gender and age distribution of patients)
6. Is training sample the CBCT scan or just the slice? Please explain.
7. What was the reason for the division into just 5 blocks (lines 101 -103)? Any supportive information for that?
8. Lines 111 - 113 are confusing. Please rewrite, and put the explanation from lines 123 - 130 earlier. It gives the contexts for the so mentioned "folders".
9. How many epochs did you train the model? What was the loss function? What were the stop criteria?
10. Image 2 should be bigger. It can hardly be seen what is on the little pictures showing the steps of CBCT processing
11. How did you train the "mini-VGG"  network? Did you use the weights from VGG16 in any way? What was the loss function used?
12. The identification of values of lambdas and overall proportional process for teeth identification should be thoroughly described. It is not sufficient just to tell that it is "form experience". What research you did? How robust is the approach, e.g. to what extent the approach is resistant to e.g. rotation of the scan? (scans not always are taken precisely in Natural Head Position). What happens when just middle tooth is only present in the block? Does the YOLO recognise properly the rectangle? Does your proportional method still work?
13. The description in lines 153-160 can hardly be understood. What are 10th and bottom 10th? Please, rewrite and explain. Maby the illustrative, supporting drawing is necessary.
Please check the text for misspellings (e. g. capitalization, comas placement, and so on). The manuscript can be considered for publication only if thoroughly rewritten and improved.

Author Response

1.The introductory part (including Related Works) would benefit from short information on the application of CBCT in disease diagnostics.

The application of CBCT

in disease diagnostics is briefly complemented to the article near Line 35.

2.All mathematical symbols used must be explained! (e. g. lambdas in eq. 1.2)

Thanks for reminding. We have made change near eqution (1)-(3) near line 174

3.The term pixel is not appropriate for 3D images as CBCT, use voxel instead.

The data we used in our study is in DICOM format which is the original information of 2D images from CBCT scan before 3D reconstruction.

4.Are the voxels cubic in shape? Please, give info in the manuscript.

The images we processed are 2D images as we expained in question 3.

5.Provide more information on the used images dataset (at least the gender and age distribution of patients)

The purpose of our study is detection of teeth from CBCT scans. Therefore, gender or age distribution of patients will not affect our results. This is the reason why the CBCT images we collected were exported with de-privacy processing.

6.Is training sample the CBCT scan or just the slice? Please explain.

The training samples are different slices from different patients. The original intention of our study is to enhance the 2D images of CBCT scan before 3D reconstruction so that we will be able to do segemention of teeth in our further study from the beginnning of image processing.

7.What was the reason for the division into just 5 blocks (lines 101 -103)? Any supportive information for that?

The division is a result of experiment. In label stage we tried different combination of blocks.  If blocks are too big, teeth belongs to other blocks are often included into box. And if the blocks are too small, there's no different between our method with one tooth one box. Then detector cannot detect blocks well when some teeth are missing. In fact, these blocks are based on the layout of human teeth. We have added this into article near 110.

8.Lines 111 - 113 are confusing. Please rewrite, and put the explanation from lines 123 - 130 earlier. It gives the contexts for the so mentioned "folders".

Thanks for reminding. Change has been applied near 122

9.How many epochs did you train the model? What was the loss function? What were the stop criteria?

We have added information you request near line 128, 136 and 164. The mini-VGG's loss function has no innovation so we just point out which common loss function we have used. On YOLO v3 we only change hyperparameters, other settings keep default.

10.Image 2 should be bigger. It can hardly be seen what is on the little pictures showing the steps of CBCT processing

Picture has been enlarged

 11.How did you train the "mini-VGG"  network? Did you use the weights from VGG16 in any way? What was the loss function used?

The mini-VGG is a VGG based method as mentioned in the article, thus it only change model structure of VGG-16 (usually only make change in module.py). Use random initialized weight, and the cross entropy loss function we use is common for classification task . Now we pointed out it in article in line 136

12.The identification of values of lambdas and overall proportional process for teeth identification should be thoroughly described. It is not sufficient just to tell that it is "form experience". What research you did? How robust is the approach, e.g. to what extent the approach is resistant to e.g. rotation of the scan? (scans not always are taken precisely in Natural Head Position). What happens when just middle tooth is only present in the block? Does the YOLO recognise properly the rectangle? Does your proportional method still work?

We have added necessary explination after your advice near comment [2]

Relavant research:

The lambdas in each block, for each tooth, on each layer class (upper and lower crown area), are different. To optimize lambdas, we chose CBCT slice from 14 patients, each on two layers. Then we initial the lambdas with estimate and run program to get tooth bounding boxes. On each picture we compare IOU of generate tooth bounding box and ground truth tooth bounding box and adjust the lambda values until the results become good enough, which is above 85%.

Robustness:

We have noticed rotation of the scan, the doctors said severed rotation would only exist when operator made a mistake and such CBCT would be abandoned. For slight rotation, our boxes are a little larger than teeth to handle this.(see figure 6a) Finally we check the method on 112 images with different rotation angle. The robustness presents in OIR metric, it shows in most cases tooth can be entirely included no matter the rotation.

For last question, Figure 7c shows when only middle tooth is only present in the block. Our method would make such mistake, and a error message will remind people to handle this.

13.The description in lines 153-160 can hardly be understood. What are 10th and bottom 10th? Please, rewrite and explain. Maby the illustrative, supporting drawing is necessary.

This paragraph is hard to understand, so we replace it by figure description on figure 5b and figure 5b is supporting drawing

Finally we appreciate it for your feedback. As the first author, this article is my first SCI article in my undergraduate stage, and inadequate sentence appears sometimes, please forgive it and relax restrictions a little more. Thanks for it, and we welcome further advice.

What's more, the article is an achievement of stage and further work is on the way. In this paper we focus on tooth detection but the entire work is based on detection+segmentation.

Revised file in attachment

Round 2

Reviewer 2 Report

The manuscript now seems to be suitable for publication.

Author Response

Dear reviewer,

Thanks for review. Now further information about CBCT voxel data and patients' age distribution has been added, in attachment PDF file

Reviewer 3 Report

Dear Authors,

I am completely satisfied with the answers and corrections you provide for my remarks 1, 2, 8, 9, 10 (great work - the picture is much better now), 11 and 12.

I am not satisfied with the answers to the following remarks. I expect the authors to revise them once again and make appropriate corrections to the manuscript.

3/4. CT scans and CBCT particularly, depict the hard tissues and (to some extent) soft tissues as a _volumes_. It is an inherent feature of the image reconstruction process. Each voxel of the CBCT scan, even from the only one slice saved as DICOM file, represents a volume. It can be treated as a result of somewhat sampling of 3D space of a patient's body in a certain coordinate system. Further, you calm you reconstructed the 3D image from the slices. How do you know the order of the appropriate slices? I suppose you had the slices indexed/numbered. The number is just the third coordinate. Now, how do you tell how much the consecutive slices are far from each other? The distance is usually equal to the height of the voxels (in other words the height of the slice). The information of it is (in most cases) in your CBCT device documentation. This information must be completed and corrected before the paper can be considered for publication.

5. The gender and particularly age information are important. In fact, there are also some other anthropological features that can influence such research. Even if you consider it not important for your research, you should clearly state it in the paper and provide any proof that it can be treated so (the best is a suitable reference). For the reader, the information gives the view that the data you used was selected to eliminate any potential factors which can influence the results. Please correct it in the manuscript by either stating clearly that it was not important (with the reference to appropriate literature) or by stating that data (CBCT scans) were collected in a way that ensures appropriate diversity.

6. Add clear information about it to the paper. The explanation is satisfactory.

7. The explanation is fine. I suggest you explain it that way in the manuscript. Lines 109 to 113 should be more descriptive. More, rephrase lines 107 - 109. Maybe using the list for enumeration would be a good idea?

13. The explanation is much better now, and no one should have a problem with following it. However, I would suggest adding the explanation to the regular text and shortening the figure subtitle.

I appreciate the work you made so far and greatly encourage you to complete the few points I mentioned. The research you did and the effort made are noticeable and definitely worth publishing.

Author Response

Dear reviewer,

Question and reply are as follow

3/4. CT scans and CBCT particularly, depict the hard tissues and (to some extent) soft tissues as a _volumes_. It is an inherent feature of the image reconstruction process. Each voxel of the CBCT scan, even from the only one slice saved as DICOM file, represents a volume. It can be treated as a result of somewhat sampling of 3D space of a patient's body in a certain coordinate system. Further, you calm you reconstructed the 3D image from the slices. How do you know the order of the appropriate slices? I suppose you had the slices indexed/numbered. The number is just the third coordinate. Now, how do you tell how much the consecutive slices are far from each other? The distance is usually equal to the height of the voxels (in other words the height of the slice). The information of it is (in most cases) in your CBCT device documentation. This information must be completed and corrected before the paper can be considered for publication.

The information of CBCT scan was provided by the device manual, such as gray density, pixel size, tube voltage and slice thickness. All the CBCT we used in our study were taken by the same operator using the same parameter regarding window size and exposure time.

The term "voxel" is modified in part 3 in the manuscript.

5. The gender and particularly age information are important. In fact, there are also some other anthropological features that can influence such research. Even if you consider it not important for your research, you should clearly state it in the paper and provide any proof that it can be treated so (the best is a suitable reference). For the reader, the information gives the view that the data you used was selected to eliminate any potential factors which can influence the results. Please correct it in the manuscript by either stating clearly that it was not important (with the reference to appropriate literature) or by stating that data (CBCT scans) were collected in a way that ensures appropriate diversity

The CBCT scan we studied were randomly chosen. The main inclusion criteria is the integrity of the dentition. The scans we chosen are with no more than 2 continous missing teeth. Therefore, we have the CT scans from both gender and an age span from 20 to 52 years.

The gender and age factors are of great importance in CBCT when studying inferior alveolar nerve(Gupta, A., Kumar, S., Singh, S. K., Kumar, A., Gupta, A., & Mehta, P. (2021). Assessment of Anterior Loop of Inferior Alveolar Nerve and Its Anatomic Variations with Age, Gender, and Dentition Status in Indian Population: A CBCT Study. International journal of dentistry2021, 1813603. ), temporomandibular joint(Yun, J. M., Choi, Y. J., Woo, S. H., & Lee, U. L. (2021). Temporomandibular joint morphology in Korean using cone-beam computed tomography: influence of age and gender. Maxillofacial plastic and reconstructive surgery43(1), 21.), maxillary sinus(Al-Zahrani, M. S., Al-Ahmari, M. M., Al-Zahrani, A. A., Al-Mutairi, K. D., & Zawawi, K. H. (2020). Prevalence and morphological variations of maxillary sinus septa in different age groups: a CBCT analysis. Annals of Saudi medicine40(3), 200–206. ) , dental pulp (Asif, M. K., Nambiar, P., Mani, S. A., Ibrahim, N. B., Khan, I. M., & Lokman, N. B. (2019). Dental age estimation in Malaysian adults based on volumetric analysis of pulp/tooth ratio using CBCT data. Legal medicine (Tokyo, Japan)36, 50–58.)or incisve canal(Linjawi, A. I., Othman, M. A., Dirham, A. A., Ghoneim, S. H., Aljohani, S. R., Dause, R. R., & A Marghalani, H. Y. (2021). Morphological evaluation of the incisive canal with reference to gender and age: A cone-beam computed tomography study. Nigerian journal of clinical practice24(11), 1596–1601.). In current stage of our study, we mainly focus on the detection and segmentation of the teeth contour.

Finally, we accept your adivce and the choosing of the CBCT data was added to Part 3.

6. Add clear information about it to the paper. The explanation is satisfactory.

(original question:

Is training sample the CBCT scan or just the slice? Please explain.

)

The explanation was added to the manuscript in Part 3.

7. The explanation is fine. I suggest you explain it that way in the manuscript. Lines 109 to 113 should be more descriptive. More, rephrase lines 107 - 109. Maybe using the list for enumeration would be a good idea?

Ok, the expliantion of this part is has more detail in line 110-131

13. The explanation is much better now, and no one should have a problem with following it. However, I would suggest adding the explanation to the regular text and shortening the figure subtitle.

We shortened Figure subtitle and edited new Regular text explanation. in line 206-212

Round 3

Reviewer 3 Report

Dear Authors,

I am impressed by the work you have done - congratulations! All my remarks were addressed and I have no more remarks. Therefore it is my pleasure to recommend the manuscript publication in its current form.